# Bringing Critical Mathematics Education and Actor–Network Theory to a Statistics Course in Mathematics Teacher Education: Actants for Articulating Complexity in Student Teachers' Foregrounds

Magnus Ödmo , Anna Chronaki * and Lisa Bjorklund Boistrup *

Department of Natural Science, Mathematics and Society, Malmö University, 211 19 Malmö, Sweden; magnus.odmo@mau.se
* Correspondence: anna.chronaki@mau.se (A.C.); lisa.bjorklund.boistrup@mau.se (L.B.B.)

**Abstract:** In this paper, we discuss how critical mathematics education (CME) and actor–network theory (ANT) come together in a mathematics teacher education course that focuses on the thematic context of climate change to study statistics. Acknowledging the complexity that student teachers encounter when asked to move from a mainly instrumental treatment of statistics toward a critical foreground of data in society, we turn to explore the actant networks, as theorized by ANT, utilized by student teachers when asked to imagine teaching from a CME perspective. For this, our study is based on a series of interviews with student teachers who participated in a statistics course where pollution data graphs were discussed, inquiring about their role as future critical mathematics teachers. The transcribed interviews, analyzed through ANT, inform us as to how student teachers' foregrounds are being shaped by actants such as the curriculum, social justice, democracy, and source critique, among others. Based on the above, we recommend that teacher education should invite active discussion of the complexity created when a CME perspective is required. This move would allow for a critical approach to critical mathematics education itself that could prepare student teachers to navigate, instead of ignoring or opposing, such complexity.

**Keywords:** statistics education; critical mathematics education (CME); climate change; actor–network theory (ANT); teacher education

## 1. Introduction

Mathematics education runs the risk of producing banal mathematical expertise, as argued by [1], and this is especially true for courses in statistics education, which tend to emphasize an instrumental use of data, disconnected from the sociopolitical and cultural contexts of their production. Moreover, this is also the effect of not considering the societal implications of certain choices when dealing with mathematics, or of referring to mathematics as something that is always neutral and objectively true. Such views and practices can be potentially dangerous in a democratic society that aims for citizens' agency in relation to societal challenges such as climate change. For this, the goal of critical mathematics education (CME) is both to empower students to become critical thinkers with respect to how mathematics is used in action, and to create awareness of such dangers [2]. It is with these thoughts in mind that we created a statistics course inspired by the principles of critical mathematics education at a teacher education program in Sweden [3]. This move aligns with the aims stated by the Swedish curriculum when it argues that school is responsible for ensuring that "each pupil on completing compulsory school can make use of critical thinking and independently formulate standpoints based on knowledge and ethical considerations" [4]. However, although critical thinking is mentioned in the Swedish curriculum, there is not much guidance as to how this should be covered. The

curriculum focuses mainly on disconnected mathematical content, without considering the need for employing thematic contexts (such as climate change) in which students' critical competencies could develop alongside mathematics. As such, the choice to cultivate a critical perspective on mathematics teaching based on climate change remains unsupported, as we have noted in a prior study on mathematics teacher education [3].

Researchers who are active in the field have investigated critical mathematics and climate change in various ways and using different approaches. For example, Steffensen et al. (2021) [5] discussed classroom lessons designed by teachers to develop students' critical mathematical competencies in a climate change context and analyzed the outcomes of these lessons. The study suggests that complex issues such as climate change bring forth an awareness of the formatting powers of mathematics. The "formatting power of mathematics", a concept developed by Skovsmose (1994) [6], signifies that mathematics potentially changes the ways in which we act, think, and experience our reality. In a related area of her research, Steffensen (2020) [7] identified how students' critical mathematical competencies appear in their attempts to enact argumentation when they discuss the themes of climate change. In the study, students participated in dialogues that involved mathematical, technological, and reflective argumentation and were based on multiple perspectives, such as environmental, economic, and ethical concerns. Her conclusion was that critical competencies are important to enable students to become critical citizens. The two studies mentioned above were empirical ones; in contrast, Hauge et al. (2017) [8] took a theoretical approach when they developed a framework that brings forward three categories that support critical reflection when mathematics is employed to discuss climate change: climate change as a vehicle, climate change as critique, and climate change as content. These categories help visualize different educational perspectives on climate change, and they can be seen as one way of grasping, and even narrowing down, the complexity when climate change is introduced into critical mathematics education. Weiland (2019) [9] took a more envisioning approach as he discussed how ideas from critical mathematics education (CME) could be used to transform the type of experience that students face with statistics in the school mathematics curriculum, and he then discussed what critical statistics education could be, using key ideas from critical mathematics literature, such as "critical thinking". Critical thinking, in the case of statistics, involves the idea of using statistics to critically examine the underlying structures and hidden assumptions present in society through specific data and, furthermore, to critique and understand these hidden assumptions. He further discussed the potential challenges faced when educators strive for critical statistics in the pedagogical context, since these often bring to the foreground sociopolitical controversies around race, sexuality, and/or ethnicity voiced within political campaigns, requiring the existing rules or norms of institutional administration and policy to be confronted.

Our way of working with climate change in this study could be conceptualized as working with a "thematic context", which offers opportunities to appreciate the potential of critical thinking in mathematics and a critical reflection on the significance of mathematics in real-life situations. The focus on "thematic contexts" was a core method for Ole Skovsmose (1994) [6] when he introduced the philosophy of critical mathematics education, with the aim of raising awareness of democratic citizenship as active participation in social practices enacted in the mathematics classroom. This was utilized by Chronaki (2000) [10] to inquire as to how mathematics teachers encounter the complexity of coordinating linkages across disciplinary areas concerning constructions of both the theme or the embedded mathematics, and the author notes the challenges encountered by teachers. Since then, several studies, such as the ones mentioned above, trying to introduce critical mathematics education in the field of institutional mathematics teaching and teacher education, have identified difficulties in the form of risks or dilemmas faced by teachers and students when they attempt to implement it within their local educational institutional settings. Moreover, they seem to agree that the teaching situation becomes even more complex when a controversial thematic context such as climate change is introduced through a

CME perspective. At this point, we can conjecture that complexity increases when new issues and potential connections are introduced into a teaching course that tries to move beyond instrumental learning of statistics. Therefore, we are interested in exploring how this complexity, an inevitable component of CME, is experienced by the student teachers themselves. For this, we employ actor–network theory (ANT) which provides a method to investigate complexity as a network of actants including the student teacher (as described in more detail below).

As such, our approach in this study is twofold: on the one hand, the philosophical standpoint of CME brings the assumption that both mathematics and mathematics education are not neutral. Skovsmose (1994) [6] discussed the concept of "the formatting power of mathematics" when he asserted that mathematics plays a role in how we see and act in the world. In other words, it produces a social and physical world after its own image. This power of mathematics is double-edged. Many great achievements in science and technology have been made possible by mathematics, but mathematics is also involved in technological catastrophes such as wars and mass destruction [11]. Mathematics not only presents the world as it is, but also formats how we act—it changes the way we think and how we perceive our physical reality. The goal of critical mathematics education is to understand this formatting power of mathematics and to empower people to examine it so that they will not be controlled by it [2]. Driven by these core ideas of CME, mathematics has been conceived as a formatting power for articulating issues of climate change [12]. Mathematics can potentially influence how climate change is perceived and formatted as solvable, predictable, etc. Coles et al. (2013) [12] presented examples of how this could be illustrated in a practice setting, and it is with these thoughts in mind that the teacher educator set up the course [3].

On the other hand, ANT states that a given social situation is made up of actants and connections comprising a network [13]. This network concept allows us to search for the actants and their connections when the student teachers enter the field of CME through the thematic context of climate change. ANT is used in this study as a methodology and philosophy to shed light on the student teachers' situation. There are several other possible theoretical frameworks that could have been used for our purpose—for instance, Foucauldian discourse analysis [14,15] or discourse theory used in mathematics education [16]. Here, we are intrigued by ANT, in that it includes an open starting point when analyzing a social situation, in the sense of not taking a certain concept of "social" for granted [13]. Rather, ANT starts from the material data, including the utterances, gestures, and objects utilized, trying to avoid fixed notions and defined concepts of the "social". Using ANT's abstract framework allows us to capture things that we otherwise might not see as they tend to remain invisible. For instance, if we were to only use CME, the utterances produced through the interviews would end up in concepts predefined by CME. However, by analyzing our interview data through the ANT approach, we do not have to rely on a CME-based conceptualization but, instead, can expand through emergent and unexpected notions.

We will describe our use of CME and ANT in more detail below, but we wish to state the purpose of this paper here, adopting concepts from these theories. The purpose of this paper was to search for complexity by identifying the actants (i.e., both human and nonhuman actors) that allow student teachers to articulate their foregrounds (i.e., expectations, aspirations, and hopes for their future) in how climate change and critical mathematics education could come together for teaching statistics. This study was organized through an obligatory statistics course and was part of a four-year teacher education program. In doing so, we were able to observe the possible tentative networks around specific actants mentioned by each of the student teachers, which reveal how they experience complexity. Complexity in our case refers to all the old and new elements, parts and connections, which are being introduced to the current and imagined teaching and learning situation, and it is this complexity (i.e., parts and connections) that we examine here.

This paper starts with this introduction in Section 1 and then continues with the theoretical considerations, where both CME and ANT are discussed in relation to the study (Section 2). The methodology section follows, describing the study's setting, including the specific methods used for data collection and data analysis (Section 3). Then, the analysis and findings are discussed in five cases of student teachers (Section 4) and, finally, the conclusions of the study are noted (Section 5).

## 2. Theoretical Considerations: CME and ANT

As mentioned above, this section discusses how the two theories (i.e., critical mathematics education (CME) and actor–network theory (ANT)) provide concepts that could support our efforts in the context of this study to delve deeper into the complexity faced by student teachers when they are asked to move from a statistics course that fulfills the curriculum to a course that utilizes the thematic context of "climate change" to approach mathematics and statistics from a critical perspective.

### 2.1. CME and Foregrounds

What is often taken as the background of a person cannot be the only factor that determines what their behavior and performance could be at a given time. Skovsmose (2007) [17] interrogated the idea of a fixed horizon of opportunities and employed the concept of foreground to refer to what new social, political, and cultural contexts might provide by arguing: "However, not the opportunities as they might exist in any socially well-defined or 'objective' form, but the opportunities as perceived by a person. Nor does the background of a person exist in any 'objective' way" (p. 6). Although the background refers to what a person has already done and experienced, such as the situations in which the person has been involved, the cultural context, and the sociopolitical context, as well as their family traditions, the person can still interpret their background in diverse ways. The foreground and the background, taken together, generate practices, perceptions, and attitudes that are regular without being consciously coordinated or governed by any rule or ritual. Moreover, the person's foreground and background need not always be in harmony with one another; they can incorporate conflicts and contradictions. A person can conceptualize different sets of foregrounds that contrast and do not align with their organized background. As such, foreground and background are continuously reworked and remolded in dynamic and relational ways with diverse characteristics. Skovsmose (2007) [17] gave a concrete example of how a person's foreground can be visualized: "an airplane passing by up there high in the sky making a fine white line, signifying that there are many different places to go" (p. 8).

Skovsmose (2007) [17] argued for the importance of grasping the specificity of an action through its intentionality. The intentions of a person are not simply grounded in their background but, equally, emerge from the way(s) in which the person revisits possibilities through action. Intentions express expectations, aspirations, and hopes. Intentions make up a constitutive part of any action. Actions become not simply caused by the past but represent forms of grasping the future. When we want to try to understand how and why a person is acting, it is important to obtain an understanding of the person's foreground and background. In this study, the concept of a person's foreground allowed us to approach student teachers, not with pre-given backgrounds that shape and sometimes fix (with stereotypes) how they act as future teachers, but with attention to how student teachers can imagine their mathematics teaching in action. As such, we turned toward investigating student teachers' visions of their roles as mathematics teachers, and focusing on their foregrounds seems useful in this respect.

Recently, Skovsmose (2023) [18] went into detail about what the core of CME is, explaining that it is about concern and hope, and arguing that with concern comes hope for change. The main concern for CME is how society is influenced by what takes place in the classroom; critical mathematics education is concerned about the students, their learning in the classroom, and their roles as future citizens. Education for citizenship

could mean preparing students to fit into the given social order, but it could also be education for autonomy, enabling students to become critical citizens. Through this, critical mathematics education is concerned about how mathematics teaching and learning addresses societal challenges, including questions about the environment [18]. Skovsmose (2023) [18] explained that traditional critical positions did not express concern about the natural environment, but only for humans' critical competence. Nature has been considered as an infinite resource that needed to be used for creating welfare. A deep concern about our environment has been broadly expressed since then. Nature has now been recognized as limited and fragile. Concern about our environment has become part of critical mathematics education. While the environment is one relevant focus for critical mathematics education, the focus on social justice is always present in CME. Environmental issues, which are the focus of this study, could also be thought of in terms of social justice. For instance, natural resources are not equally distributed around the world. Some groups of people benefit much more than others. Some nations use far more resources per citizen than other nations. Pollution is a significant problem, but it does not affect everybody in the same ways. Skovsmose (2023) [18] argued for a mathematics education for social justice in terms of processes that engage students in the very formulation of what social justice could mean, and not as an education informing students about justices and injustices. A principal step is to engage students and student teachers in a pedagogical process of identifying and articulating for themselves and their community what they find to be just or unjust, and to be ready to critically challenge fixed opinions. We have fully adopted this standpoint in our study and implemented it in workshops with student teachers, where we aimed not to lecture about what justice and injustice are, but to invite the student teachers to relate with specific examples where they could foreground their potential actions. We also made use of the model presented by Skovsmose and Borba (2004) [19], which conceptualizes the interplay amongst the current situation (CS), the imagined situation (IS), and the arranged situation (AS), as exemplified by the corners of the triangle in Figure 1.

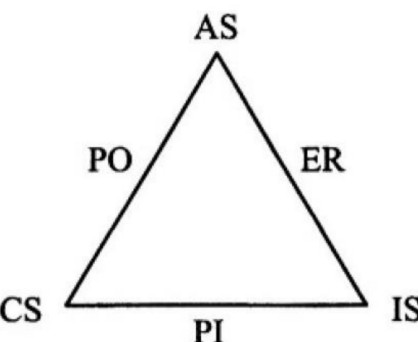

**Figure 1.** Model of researching critical mathematics education in action.

This study focuses on the imagined situation (IS) as the desired pedagogical setting, where student teachers can articulate their foregrounds as critical mathematics teachers through a statistics course that utilizes climate change as a thematic context. To move toward the imagined situation, the student teachers must experience a process of change from their current situation (Figure 1). For this process, developing spaces for pedagogical imagination (PI) is important to foster the relationship between the current situation and the imagined situation. In this, the relationship between the current situation and the arranged situation is established by practical organization (PO). Practical organization consists of planning activities perceived as necessary for establishing an arranged situation. Finally, explorative reasoning (ER) refers to the critical and analytical process of reconsidering the pros and cons of the imagined situation considering participants' experiences and their roles in the arranged situation. Thus, explorative reasoning is a process through which the feasibility of pedagogical imagination is discussed, along with the innovative elements that allow for its practical organization [19] (p. 216). We used this model to construct our

questions in the interview guide (Appendix A): one question that deals with the arranged situation in the course, two questions that concern the imagined situation, and two open questions that allow the student teachers to refer to the current situation. More on the interview questions will follow in the Methodology section below.

### 2.2. ANT and Actants

CME and foregrounds do not seem sufficient to understand what enables and prevents student teachers from changing the current situation to an imagined situation that foregrounds the critical mathematics education vision. Vithal (2000) [20] argued that CME in fact introduces a serious contradiction when exploring a theory that attempts to introduce a critical democratic perspective to an educational setting without it being an imposition. However, that imposition must be made ". . .in the hypothetical situation precisely in order to make such ideas more widely available and to understand what they can mean in reality" [20] (p. 6). This contradiction could be one reason why student teachers are prevented from changing the current situation to an imagined situation; in order to find other issues like this one, we need to ask what enables the student teachers to act or not to act. Actor–network theory (ANT) provides a way of describing this affordance through a network [13,21]. ANT tries to understand the situation based on relationships between what are called actors or actants. The difference between actors and actants is that the term actant refers to an abstract structure, whereas the term actor is a concrete one, as described by Latour ". . .going from abstract structure -actants- to concrete ones -actors" [22] (p. 8). The term actant is used to suggest that agency is assigned not only to humans but also to nonhumans such as animals, physical things, and ideas. In this study, we adopted the term actant throughout the investigation.

As such, the actant represents anything that has the possibility of producing a particular effect and, thus, has agency (Smelser and Baltes, 2001) [23], or in Latour's words "An actant can literally be anything provided it is granted to be the source of an action" [23] (p. 7). Actants can also be socially constructed ideas, such as legal codes and ideologies [24]. The relationships between actants that we iterate as a collective over time represent a way of thinking about how things are also in a process of re-creation in critical mathematics education [25]. Furthermore, there are multiple connections amongst actants that constitute a network, and zooming into a particular actant would reveal yet another network [13]. Specifically, Latour argues "A network, in this second meaning of the word, is more like what you record through a Geiger counter that clicks every time a new element, invisible before, has been made visible to the inquirer" [26] (p. 799). To give a concrete example, one could consider a classroom with a projector. During a lecture, that projector would constitute an actant, since it is a source of actions in how the social situation plays out. With actor–network theory, when one tries to describe the situation, one would have to include the projector as an actant, along with other sources that allow action. However, if the projector breaks down, then there might be a need to zoom into "the projector" and reveal the network that describes the situation, (its parts, wires, etc.), but as long as the projector is working, for the purposes of describing the situation, the actant "the projector" is sufficient [27].

In this, agency can be described as the ability of an actant to mediate another actant. In ANT, there are two main concepts: mediators and intermediaries. Mediators ". . .transform, translate, distort, and modify the meaning of the elements they are supposed to carry" [13] (p. 39). Intermediaries, on the other hand, are what transport meaning without transformation: ". . .defining its inputs is enough to define its outputs" (p. 39). An example of an intermediary can be an administrative instance that handles some applications and simply transfers them to the next instance. Mediators, on the other hand, transform meaning and content. However, how do we distinguish between mediators and intermediaries? Latour mentions that to use ANT is nothing more than to ". . .become sensitive to the differences in the literary, scientific, moral, political, and empirical dimensions of the two types of accounts" (p. 109). This means that, in our case, when inquiring into how student teachers

talk about their foregrounds as future critical teachers of statistics, we must be sensitive to when something is introduced that transforms meaning. This means that we encounter entities that, for some other investigations, could be considered actants, but not for a particular one and, therefore, can be ignored. ANT emphasizes language displacement from one frame of reference to the next that Latour (2005) [13] calls infralanguage and argues "In my experience, this is a better way for the vocabulary of the actors to be heard loud and clear" (p. 30). An example of how we use this can be found in the Methodology section.

The notions of actant and network allow us to conceptualize a picture of the complexity that a student teacher in teacher education moves with when engaging with critical mathematics education and climate change as a process and not as a fixed outcome. This allows us to revisit critical mathematics education as a process where mathematics, as a nonhuman actant, also acts and ask questions such as "What is mathematics? What is it that we do when we do mathematics? Who acts? And with whom? Is it us? Only us? Us alone?" [25] (p. 31). Such questions allow us to move beyond the immediate concern of mathematics as a human construct and explore the potential network of relations afforded across diverse actants. Latour (2011) [26] explained how both actants and networks suggest fragility, and the empty spaces between the arrays imply possibilities, as they could possibly be inhabited by another actant or connections between present actants. Especially important is what the network does to universality; any part of the network is accessible from anywhere in the network, just as there are enough "...antennas, relays, repeaters, and so on," [26] (p. 802) to sustain the network. Latour further argued that "In network, it's the work that is becoming foregrounded, and this is why some suggest using the word worknet instead" [26] (p. 802). In other words, by using the concept of a network, it is possible to localize where and through which other actants a given actant is influenced. Rather than an existing stable entity, the network can better assign a mode of inquiry that "...learns to list, at the occasion of a trial, the unexpected beings necessary for any entity to exist" [26] (p. 799). Networks make visible the configuration in which actants—human and nonhuman—are entangled and, in different ways, emerge as significant and powerful. Latour wants us to move away from accepted concepts that hinder deeper understanding, for example "...nature, society, or power, notions that before were able to expand mysteriously everywhere at no cost" [26] (p. 802). In so doing, he writes, the forces that affect people can be seen in a clearer way. These ways of working with actants and their interconnections enable us to articulate networks of specific situated relations.

With these ideas in mind, we can now formulate our research question: what might be the networks of relations that transform the mathematics student teachers' attempts to foreground classroom teaching within the milieu of CME through the thematic context of climate change? To explore this complex question, we undertook an empirical investigation in which we inquired as to the actants and their interconnections and relationships, as presented in a series of interviews with student teachers after completing a statistics course for critical mathematics education and climate change. ANT does not provide much explanation about *why* the actants and their interconnections are there, or about how they contribute towards creating a relational network that affords student teachers a critical stance for both statistics and climate change. In this, specific concepts of CME, such as foreground, imagined situation, and pedagogical imagination, could allow us to answer such questions.

The present study can be viewed as being situated in the above model (Figure 1), where the student teachers participate as co-researchers in this complex process, offering vital ideas about how their foregrounds as critical mathematics teachers of statistics could be materialized (or not) through the thematic context of climate change. Since the student teachers' ideas are heavily grounded in both human and nonhuman resources related to their living experiences, our study turns to the concept of actants as discussed in actor–network theory (ANT) as a way to inquire as to the types of knowledge, and elements that act with them, to reach their imagined situation.

## 3. Methodology

In this section, we outline the methodology of this study by recounting how the theories have contributed to its organization around the research question. We also reflect on the challenges when these theories are utilized with empirical data and discuss how these could be overcome. The present section comprises four subsections: the first discusses the study setting by providing details of the statistics course in teacher education around the vision of critical mathematics education through the thematic context of climate change; the second outlines methods for data collection; the third outlines the methods used for data analysis; and the fourth discusses the ethics of this study.

### 3.1. Study Setting: A Statistics Course for Critical Mathematics Teacher Education

As mentioned in the introduction, this study took place at a teacher education program at a university in Sweden, focusing on a compulsory statistics course for student teachers. It contained five cycles of two-hour lectures and two-hour workshops. The first author of this paper was the teacher educator for the course in this study, whilst the overall design and analysis were the collaborative work of all of the authors. In this sense, this paper could be seen as the extension of a prior self-study (see also: [3]) where the second and third authors, as critical collaborators, participated actively in both studies.

The course typically comprises 20 to 40 students studying to become teachers for pupils in the age group of 10 to 12 years old. The statistics course deals with fundamental concepts such as mean and median values, as well as methods for data visualization such as diagrams, graphs, histograms, table charts, etc. The course also examines some didactic ideas that can be used in a school setting. Being sensitized to the need to revisit mathematics teacher education in the light of current societal and environmental urgencies, the present study considers both climate change and critical mathematics education as key axes for redesigning the statistics course (removed for peer review). This is exemplified during the course by a short introduction to what CME is, including related activities provided during the workshops. In these activities, we asked the student teachers to reflect on and discuss how different types of graphs could change the perception of climate change. For this, the graphs identified already by Coles et al. (2013) [12] were employed (Figure 2). The choice of these graphs was made on the basis that, in our opinion, they clearly bring forth the formatting properties of mathematics and are suitable since they deal with climate change, our thematic context of choice. Although Coles et al. (2013) [12] proposed these graphs for use in secondary school classrooms, in this study we employed them with student teachers enrolled on a statistics course. The first two graphs below show the amounts of carbon dioxide emitted from the countries Great Britain and India from the year 1880 until 2008. The first graph shows the year-to-year emissions from both countries. The second is a cumulative graph, where the previous years' emissions are added to the coming year's emissions. As an example, reading the data point from the year 2008 shows the total amount of emissions since 1880 for that country. The third graph shows the emissions per capita from 1950 to 2008 for the two countries.

### 3.2. The Interview as a Method for Eliciting Student Teachers' Foregrounds

Our research question, as mentioned above, was "What might be the networks of relations that transform the mathematics student teachers' attempts to foreground class-room teaching within the milieu of CME through the thematic context of climate change?" To explore this question, we conducted an inquiry study. Having organized the statistics course around the two key axes of climate change and critical mathematics, we moved toward organizing our inquiry based on interviews with a small number of student teachers who volunteered to participate. The data for the present study were derived from a series of semi-structured interviews with five student teachers.

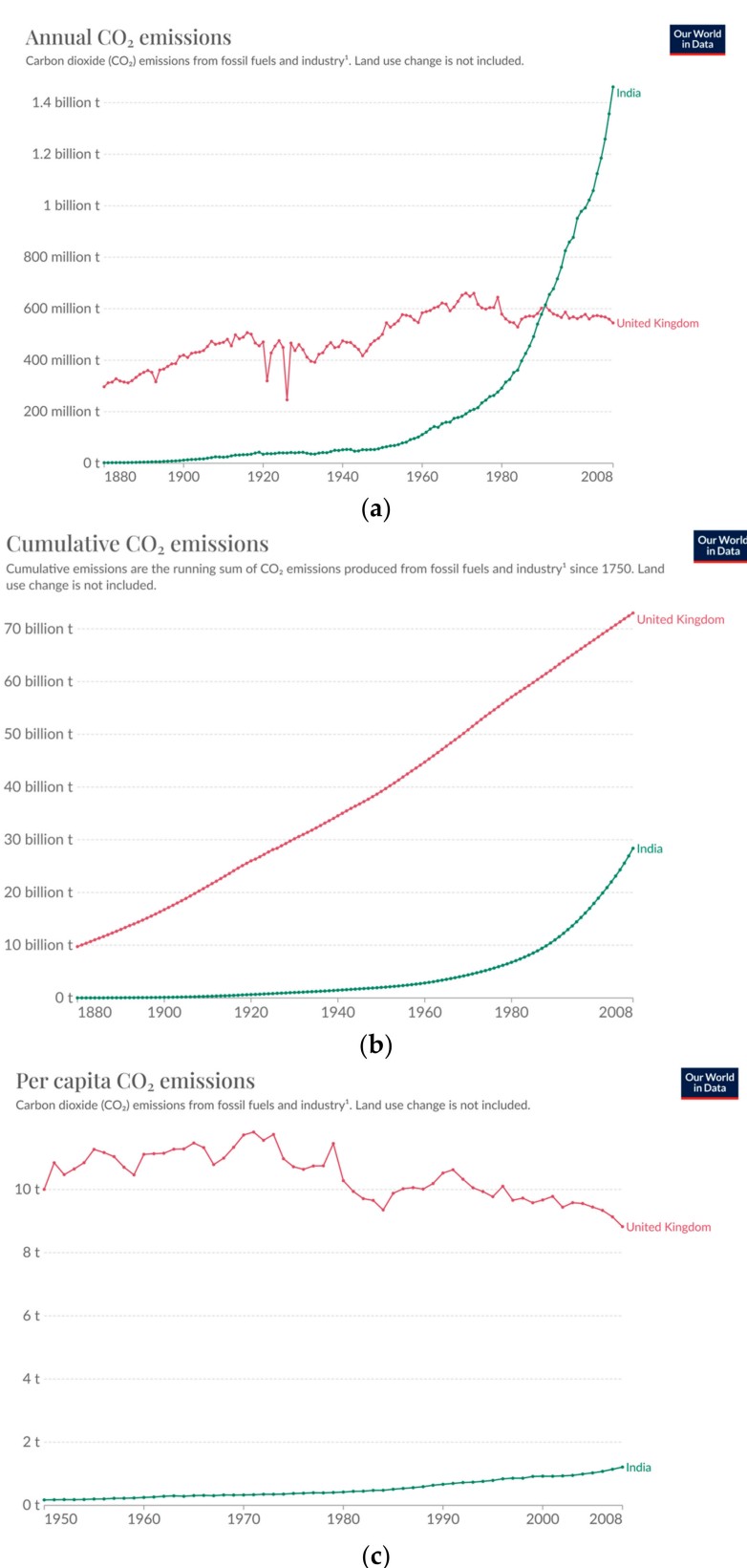

**Figure 2.** Graphs used for the CME statistics course (graphs created at https://ourworldindata.org/, source: Coles et al., 2013 [12], pp. 45–46). (**a**) Annual fossil fuel emissions for the UK and India (1880–2008). (**b**) Cumulative fossil fuel emissions for the UK and India (1880–2008). (**c**) Annual fossil fuel emissions per capita for UK and India (1950–2008).

An important principle from Brinkmann and Kvale (2018) [28] used in the interviews is to allow silence. By allowing for pauses in the conversation, the student teachers are given time to associate and reflect, and then they can break the silence themselves with significant information. These pauses are denoted as # for shorter pauses and ## for longer pauses in the transcripts. The interviews were approximately 30 min long, and an interview guide (Appendix A) was used, containing five main axes of questions around which the conversation could take place. The questions were as follows: Q1: What did you think of the statistics course? (A simple, open question to start the conversation.) Q2: In the lectures I mentioned the following examples: How did your thoughts go during the lecture and afterwards, around this example? (Appendix A) Q3: How could you as a teacher work with this in your teaching in a class, grades 4–6? Q4: How would a class in grades 4–6 benefit from this in their teaching? Q5: Is there anything else you want to add?

For each question, there were supplementary or follow-up questions addressing the guidelines by Brinkmann and Kvale (2018) [28] for how a semi-structured interview should be conducted. According to them, the first question should be an open question that aims at producing spontaneous, rich descriptions where the student teachers themselves describe what they experience as the main aspects of what is being investigated. The other four questions aim at being easy to understand, short, and without academic language. The questions were also evaluated with respect to both thematic and dynamic dimensions: thematically about producing knowledge, and dynamically regarding the interpersonal relationship in the interview. We aimed at formulating the questions in such a way that they would contribute both thematically to knowledge production and dynamically by promoting good interview interactions. The follow-up questions aimed at prolonging the student teachers' answers and maintaining a curious, persistent, and critical attitude during the interviews [28]. "Could you tell me more about that?" is an example of a follow-up question that is used—a kind of a probing question with the aim of pursuing the answers and probing their contents, but without stating which dimensions are to be taken into consideration. Structuring questions is another example.

The aim of this type of question is to steer the course of the interview in order to break off long answers that are not the focus to the investigation—for example, by briefly stating the understanding of an answer, and then saying "I feel that you see many benefits regarding the use of climate change as a subject, but how do you think it affects the learning of mathematics?". A concrete example of this used during the interviews is the question "How do you think it affects the learning of mathematics or the understanding of mathematics?", used when the course of the interview tended to go in an unwanted direction. Another example is "The way we're talking about these diagrams now, do you think that talking about it in this way is something that benefits students' math learning?". Further, the questions were produced using the research methodology (Figure 1) presented in the Theoretical Considerations section. Question 1 (Appendix A) is an open question that allows the student teachers to reflect on all parts of the model (Figure 1), i.e., current situation, arranged situation, and imagined situation. Question 2 revolves around the arranged situation, starting with a description and a reiteration of what was worked on during the workshops. Questions 3 and 4 focus on the imagined situation, and a final question asks whether the student teacher wishes to add anything.

Järvinen and Mik-Meyer (2020) [29] discussed eight different approaches to qualitative analysis, of which ANT was one. They argued that the main objection to using qualitative interviews together with ANT is that the interview contradicts the fundamental premise of ANT, i.e., that human actors should not be privileged over nonhuman actors. Interviews may vary in how much focus they place on the key human actors, i.e., the interviewee and interviewer. However, over the last decade or so, several studies have applied ANT concepts to interview materials, with a focus on mapping the interactions between human and nonhuman actors [30–35]. Specifically, Järvinen and Mik-Meyer (2020) [29] elaborated on challenges in ANT-inspired analyses of interview materials. Firstly, it is important that the interview opens a space where the interviewees are given the opportunity to articulate

those elements in the network that are important to them. The interview therefore needs to adopt an open and exploratory approach. This was achieved in our study by making sure that the questions and the follow-up questions did not close this space but promoted further elaborations on the same topic. One of sociology's core tasks is to show how social structures operate in relation to human actors, and since ANT has a different starting point it leads us to another challenge in ANT analyses, concerning how we view the data material and what the informants tell us. When we use the principle of "following the actor", we cannot presume that social structures exist and exert influence on the actors. This must remain an open, empirical question. We must take the informants' statements at face value instead of explaining their statements with reference to factors outside of the data. We should not assume that social categories such as "ethnic minority", "working class", or "gender" are necessarily suitable for understanding a given situation. An ANT analysis requires that the relevance of these and similar categories is shown through the data—that these categories "act" in the data—before they can be assigned relevance in the analysis.

All interviews were performed in the Swedish language, recorded, transcribed in Swedish, and then translated from Swedish to English. The analysis involved carefully reading all interviews as data produced by student teachers as they responded to the interview questions. The data were also analyzed with respect to our research question, which means that we focused our analysis on whether or how the student teachers related to their imagined future teaching and classroom, i.e., their "foreground". This occurred mostly in their responses to questions 3 and 4 from our interview guide (Appendix A). These questions are directly related to the concept of foreground and the imagined situation. However, there were instances based on questions 1 and 2 where this also occurred. This dataset is what we analyze in Section 3.3.

### 3.3. Analyzing the Interviews: Inquiring for Actants in Student Teachers' Foregrounds

The analysis here was also inspired by the study of Boistrup and Valero (in press) [36], especially with regards to how the data were handled. They embraced some of the notions and analytical strategies of Bruno Latour to think about the narratives of mathematics education as a field of research. This allowed them to conceptualize a network in which mathematics education forms a part. Working with Latourian tools, they performed a limited empirical investigation of how mathematics education research texts from 2004 to 2020 establish relationships to PISA, as well as which controversies are noticeable in the research. A noticeable difference is that while they analyzed documents, we analyzed interviews. However, the methodology of how actants can be located is similar, using one actant as the entry point, finding connections to other actants, and then organizing these actants in a spreadsheet. In addition, the frequencies of actants were located in the interviews for each of the student teachers (Appendix C).

Finally, the digital tool Visio was used to create the tentative actant networks (as seen in Section 4), allowing us to move the actants around without losing the connections. The networks were created by first drawing the actant student teacher, since that was our entry point into the investigation, followed by the generalized actants and the connections. We continued this process until we had covered the identified generalized actants and connections from the student teachers' utterances. The layout of the different actants was made on a practical basis to allow all of the connections to be seen as clearly as possible. This layout can also be achieved in other ways; what is constant is the connections between the actants.

We now apply our concept of actants and connections from ANT, as described in the theoretical section. In the dataset, key actants were identified and systematically organized in a spreadsheet using the categories of actants, generalized actants, description of generalized actants, and connections. This spreadsheet is not included in the article, due to its large size. First, by carefully reading through the student teachers' statements, we noticed when things were introduced in their utterances that transformed meaning. The word or words used to describe that "something" were then transferred to the spreadsheet

in the actant column. As an example, one of the student teachers talked about CME and then related it to the idea of source criticism; we noticed from their statement that the student teacher then started to talk about imagining CME exercises as going through newspapers to search for statistical content and critically examine them. The words "source criticism" were then transferred to the actant column.

Second, the generalized actant denotes when similar terms are used in the data for the same thing. For instance, if the student teachers talked about "being fooled" or "being tricked", we thematically grouped this under the generalized actant "manipulation". In this way, we used ANT's notion of an infralanguage [13], making sure that as little transformation as possible of the intended meaning had taken place, to allow us to see the actant more clearly. Third, the description of generalized actants gives a more in-depth description of the generalized actant than the name provides. Fourth, "connections" specify what inter-actant relations the student teacher creates through their utterances. For instance, in the example above, where the student teacher connected CME with source criticism, the connection is denoted in the column "connections" as "CME-source criticism". Overall, our inquiry envisions student teachers as potential actants (i.e., future critical math teachers) through a network of other actants and interconnections. This means that when trying to map the network, i.e., the actants and their connections, we imagined each of the student teachers as entry points. In Appendix B, we list the generalized actants for all five student teachers.

These generalized actants represent the actants that were mentioned in the interviews in relation to how the student teachers foreground themselves as critical statistics teachers. The generalized actants are illustrated by quotes from the interviews, and descriptions of the generalized actants are provided. An example from the list in Appendix B is the generalized actant "Source criticism". In the description, we can see that this is about the "The idea that facts need to be checked for accuracy". The example quote is from the interview with Nadir when she said: "There is so much information online, but it must be from sources that are close to our time and so that you have discussions like this about source criticism". Utilizing the generalized actants and connections from our spreadsheet, we can now create a network for each of the student teachers based on their utterances. This allows us to easily see which generalized actants each student teacher connects to, as well as which generalized actants they make connections between.

Our identified generalized actants allowed us to discern the networks described by the student teachers when entering the field of CME in the thematic context of climate change, in relation to their foreground. We argue that these generalized actants influence the student teachers' foregrounds, i.e., their hopes and aspirations for their future teaching in statistics using CME and climate change. The generalized actants provide constraints and affordances (Appendix B): constraints in the sense that the actant suggests certain ways of doing things and not others, and affordances as the actants provide sources of actions.

*3.4. Ethical Considerations*

The interviews were conducted by the first author of this article and took place after the statistics course was already completed and assessed. This was to reduce the risk of any ethical implications or bias from the fact that the interviewer was also the teacher of the course. Otherwise, there could be a risk of student teachers thinking that their participation would somehow influence their grading on the course. Although all the student teachers who participated in the course were given the chance to participate in the follow-up interview, five of them agreed, and they were all female. This might be partially because females are over-represented in this statistics course (i.e., at time of this study there were 19 females out of the 26 students enrolled), but this certainly needs further analysis considering the increased feminization of the teaching profession. The fact that the teacher was also the interviewer might have affected who and how many participated in the interview. It is hard to say exactly how and in what way(s) this influenced participation. As we see it, it could go both ways; it could mean fewer participants, but it could also

mean more participants than would otherwise have been the case. For instance, having some general ideas about who the interviewer is and what they stand for could remove some uncertainty that might cause some student teachers not to participate. On the other hand, a bad experience with the course and the teacher in general might cause reluctance to participate. The teacher being the interviewer could also potentially influence the ways in which student teachers answer the questions, consciously or unconsciously giving the answers that they think they know that the teachers want to hear. To deal with this, it was important to emphasize that the course was completed, and that the interviewer no longer had the role of the teacher as evaluator of the learning process. In this case, it could also go either way, because now the student teachers had the chance to say exactly what they thought about critical mathematics, without the risk of being judged or graded for it. Based on the interviews, there were some cases where this happened, which we interpreted as a sign of the participants feeling free to express their thoughts.

All the interview participants were ensured that they would remain anonymous and that none of their personal information would be accessed. As such, in our interview setting, we stayed away from personal questions that could make any direct reference to their background or personal history. This was for ethical reasons, making sure that it would not be possible to use the information provided by the informants to figure out their identity. All the participants have been given pseudonyms in this text (i.e., Nicole, Sophie, Nadir, Iman, and Estelle). Conversation was initiated with the local ethical authorities at the university where we are active, and we received a letter of confirmation stating that no formal vetting application to the "Swedish ethical review authority" was needed due to the nature of the study.

## 4. Actant Networks for Student Teachers' Foregrounds

The actants and actant networks created by the student teachers as described below should be seen as a tentative answer to our inquiry addressing the following question: What might be the networks of actants and their connections that articulate the mathematics student teachers' attempts to foreground classroom teaching within the milieu of CME through the thematic context of climate change? Along each of the visualized actant networks, we give a short summary of the interview, followed by some example quotes. We investigated these quotes using concepts from CME and ANT and situated how the student teachers brought the actants into relation with the core elements of bringing CME into action (see the model in Figure 1, Section 2). Many of their utterances, produced through the interviews, can be situated in how the student teachers' imagined situation is foregrounded as they try to express and articulate specific ways of getting there that involve them in explorative reasoning (ER) and pedagogical imagination (PI). This is partly due to the nature of the questions asked in the interviews, which place an emphasis on their future classroom and teaching. The imagined situation, we argue, is a part of the student teacher's foreground, as it grounds their hopes and aspirations in relation to specific materializations concerning the teaching and learning situation. The imagined situation is often set in contrast, in the student teachers' utterances, to the current situation (CS), where mathematics is of a more instrumental type. With this in mind, we present below the cases of all five student teachers and discuss the actant networks that interpret each of their foregrounds as critical mathematics teachers employing the thematic context of climate change in their teaching of statistics. For clarity, for the first student teacher, Nicole, we have put the words stemming from the theoretical framework in italics.

### 4.1. The Case of Nicole: Mathematics as a Social Activity and Pupils as Not Manipulated

As we can see in Figure 3, the actants comprising Nicole's network are "manipulation", "climate change", "real world", "math as social activity", and "fear and hope".

All of the actants in Nicole's actant network (Figure 3) are directly connected to CME, but not to other actants. Nicole talked very much about CME as a social activity, and she referred to her own experiences of acting as playing and making something physical

during the class. She recollected "*I think you learn more when you sort of get to do something physically as well. For me, if I think back to my own schooling—I remember those things where we kind of had plays. But I don't remember all the written tests I had*", and she added "*Then I also think it will be a lot of fun, so that's one reason they'll remember it more*". Nicole connected CME (*the arranged situation*) with the *actant* "math as social activity", resorting to her past experiences for conceptualizing what CME might be. Then, by using *pedagogical imagination*, she tried to envision what the benefits would be by using CME and "math as social activity" together in that *imagined situation*. She also connected the *actant* "math as social activity" with the actant "fear and hope", in that she hoped, based on her past experiences, that it could enhance the ability of students to remember what was said during the teaching and learning situation.

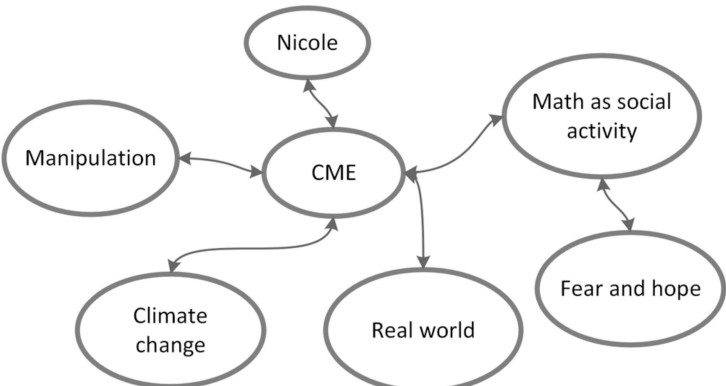

**Figure 3.** Nicole's actant network.

The *connection* she made between CME and "math as social activity", we argue, allowed her to make the next step in her *foregrounding* of the *imagined situation*. As she also said, "*You can manipulate quite a lot with diagrams depending on what you choose to focus on and what purpose you have with the various diagrams, sort of*". Nicole connected CME with the actant "manipulation". Nicole wanted to make sure that the pupils are not tricked or manipulated when it comes to the content of statistics. We can see that Nicole's *foreground* and role as a future teacher in relation to CME and climate change is to *facilitate social activity and make sure the pupils are not manipulated*.

### 4.2. The Case of Sophie: The Ethics and Source Criticism

As we can see in Figure 4, the actants that comprised Sophie's network around CME are "pupils' age", "source criticism", "math as social activity", "climate change", "politics", and "social justice".

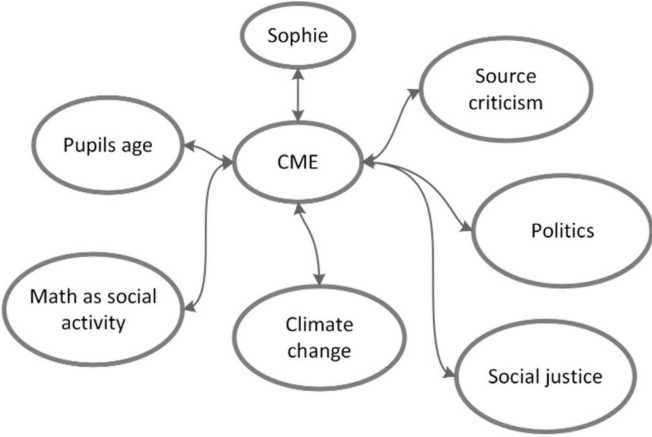

**Figure 4.** Sophie's actant network.

All of the actants in Sophie's actant network (Figure 4) are connected directly to CME, but not amongst themselves. Sophie talked about CME and climate change as she reasoned about how children are the future and, therefore, climate change should be especially important to them. Sophie embraced the idea of classroom argumentation and reasoning in the classroom to discuss the offered diagrams concerning India and Great Britain, so to highlight their underlying meanings she said "*Two different diagrams, the same countries but different diagrams, it can be a bit of source criticism too. What is actually right?*" Sophie connected CME in the arranged situation with the actant "source criticism" in her way of conceptualizing CME. In the interview, she said:

> "...*and it's good if you had seen this in a newspaper that you would have been able to understand it, because we are discussing if two students had discussed then they would have been able to learn about diagrams together and then be able to understand it themselves, if they had seen it in a newspaper or elsewhere*".

By using pedagogical imagination, she gave a concrete example of when this would be useful for understanding diagrams when reading newspapers. In response to the question "Do you have any more situations like this where it might be useful to have that understanding?", she responded "*So also on social media, if there is, so they take up statistics on how, who has the most followers. I don't know, even the weather*". Unfolding her pedagogical imagination further, she gave another concrete example that could be used in her imagined situation: students discussing statistics on social media or concerning weather. We argue that the connection she made between CME and the actant "source criticism" is what enabled her, in her pedagogical imagination, to come up with the idea of investigating statistics on social media or weather data. All in all, Sophie's foreground and role as a future teacher in relation to CME and climate change is to *go into ethical discussion in the classroom and see CME as an instance of source criticism*.

### 4.3. The Case of Nadir: A Political Stance for Democracy through Discussion

In Figure 5 we can see that the actants encountered by Nadir are "democracy", "politics", "interdisciplinary", "real world", "politics", "CME", "instrumental math", "social justice", "climate change", "math as social activity", "UN", "pupils' age'", "fear and hope", and "source criticism".

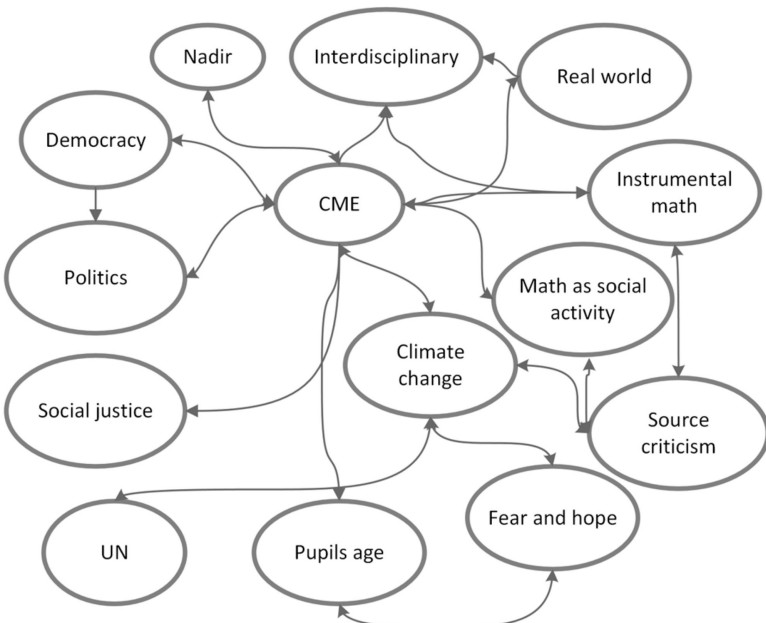

**Figure 5.** Nadir's actant network.

In the actant network in Figure 5, one can see many more actants as compared to the previous two cases of Nicole and Sophie, with many more complex connections across and between them. Nadir went into lengthy and complex thoughts in relation to CME and climate change. She took a more political perspective and connected democracy and social justice to CME, even relating these ideas to how the United Nations addresses climate change. Nadir related to our current reality of climate crisis and emphasized the importance of facts and source criticism. She compared how democratic countries respond to issues of climate change and expressed a desire to see serious discussions around climate change in countries that are assumed to be nondemocratic. For instance, she said:

> *"Politicians have a lot, really decide like this, this international, or this national, within our country, what shall apply, what is done. That's not how it looks in all countries, really. Especially if it's not a democracy."*

Nadir connected CME (the arranged situation) with the actant "democracy". She foregrounded herself as investigating what CME is in relation to democracy. Further on in the interview, by using pedagogical imagination, she came up with the idea of using emissions data (for example, from China) to discuss whether emission levels are related to whether the country is a democracy. The connection Nadir made between the actants "CME" and "democracy", we argue, enabled her to construct a pedagogical imagination to reach a concrete example to examine. Nadir further connected CME with interdisciplinary study, and she argued that since subject areas are not separated in the real world, the school curriculum should not separate them either. She said:

> *"They [pupils] must be able to make logical arguments like this and be able to respond to arguments, there are certain, so like this, knowledge requirements, it's not just that they should know exactly what a table is, but it will be, so it's [real-world] not as separated as it [school] is."*

In relation to this connection to the real world, Nadir was also concerned about the pupils' feelings in relation to climate change, as exemplified in this statement:

> *"It should not be too negative so that, that the students get too negative a view, there is hope there is like, present it that way, but at the same time they need to be aware of this stuff."*

CME and "real world" were connected actants for Nadir, who also connected the actant "fear and hope" with the real world, in the sense that she was concerned that CME using climate change as a thematic context might inflict negative feelings. We argue that the actant "real world" both improves and restricts her ability to foreground herself as a critical mathematics teacher. The actant "real world" presents her with a dilemma. Here, we can see teacher education playing a vital role in resolving and clarifying these types of dilemmas. Nadir's foreground/role as a future teacher in relation to CME and climate change is mostly a *political one, emphasizing the democratic importance of discussions*. She also brought up the UN as a part of setting up her future teaching, as the UN provides guidelines in relation to education and climate change. Nadir was also concerned with the emotional impact of introducing climate change and how best to deal with this.

*4.4. The Case of Iman: Grounded in the Curriculum for Pupils*

In Figure 6 we can see that the actants involved in Iman's network are "instrumental statistics", "manipulation", "interdisciplinary", "math as social activity", "CME", "teacher traditions", "curriculum", "climate change", "social justice", "fear and hope", "real world", and "democracy".

Again, we can see a complex actant network (Figure 6) with many interconnections established. Iman also expressed complex thoughts about CME through the thematic context of climate change, much like Nadir did, but Iman tried to turn to the curriculum to resolve her issues in relation to CME. Iman connected social justice with CME and connected climate change with the curriculum. She also reasoned around how the curriculum could

be interpreted in many ways, and she suggested choosing based on one's own knowledge. For instance, she said:

> "The curriculum can be twisted and turned so that you align with it, if you take up the broader issues, I think. So, you can always find something that matches more or less well. I think that you yourself have to be committed or knowledgeable about the field, otherwise it will just be a mess if you try to lecture about something you don't know."

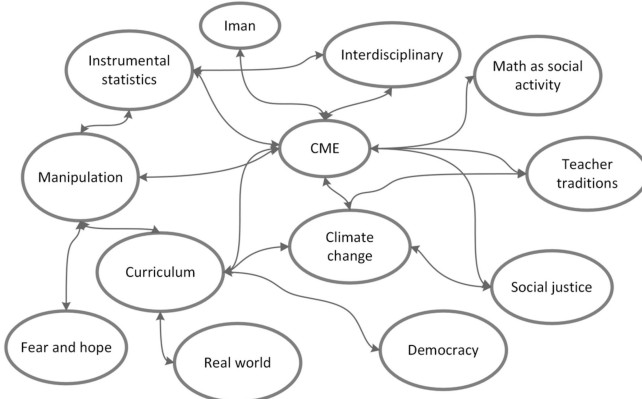

**Figure 6.** Iman's actant network.

Iman connected the actants "CME" and "curriculum" in her efforts to foreground her imagined situation as a critical mathematics teacher. Her statement can be seen as a part of the process of explorative reasoning (Figure 1) towards the imagined situation. We argue that the actant "curriculum" both affords her opportunities and restricts her in her quest to reach the imagined situation. While the curriculum states that critical thinking should be practiced, it does not say how and to what extent it should be covered; this could potentially be a source of dilemma for the teacher, as mentioned earlier. In her exploratory reasoning, she also found a way of overcoming this obstacle, arguing that parts of the curriculum could be seen as covered in a broader sense. Whether this argument should be used to introduce critical thinking is an open question. On the one hand, she based her choice of using CME and climate change on the curriculum, but she preferred to look at it from the pupil's perspective. She argued that she and her future pupils should not be easily manipulated. She also said:

> "And when they usually ask why we should learn this, well it's so you won't be fooled by someone else or something else. It's something they can understand, they don't want to feel stupid, nobody wants to feel stupid."

Iman connected the actants "CME" and "manipulation" when she tried to foreground her imagined situation, and she also connected the actants "manipulation" and "fear and hope". The connection that she made between "CME" and "manipulation" allowed her to use her pedagogical imagination to find the argument that she would use with the pupils to persuade them to participate, at the same time presenting hope that they would not feel stupid, but also fear that not participating could result in them being perceived as stupid. Iman also reasoned about teaching traditions in relation to CME, considering whether to push issues or to allow everything in the discussions—for instance, allowing climate change denial. Her foreground and role as a future teacher in relation to CME and climate change is *grounded in the curriculum and concerned with pupils not being manipulated*, and she sees opportunities to talk both about climate change and social justice.

*4.5. The Case of Estelle: Caring for Young Pupils' Thinking and Feeling*

In Figure 7 we can see that the actants in Estelle's network are "pupils' age", "CME", "math as social activity", "climate change", "real world", "instrumental statistics", "fear and hope", and "manipulation".

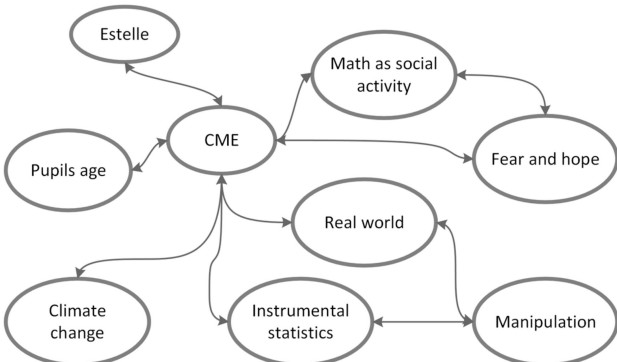

**Figure 7.** Estelle's actant network.

At first glance of the actants in the network (Figure 7), it looks as though they are mostly connected to the central actant "CME", as in the cases of Nicole and Sophie, with only two subnetworks created: one between "math as social activity" and "fear and hope", and the other amongst "instrumental statistics", "real world", and "manipulation". Estelle explicitly stated that she actively considers her role as a future teacher in statistics, along with how she could incorporate CME and climate change. She said:

*"I think that at that age there are so many other things that are more important, and math has generally been a subject that many people find difficult. That oh, and then it becomes extra important that, well, they should be able to relate to it and see what is around them in order to show this with carbon dioxide, you can certainly find ways to show the emissions but perhaps on a smaller level, where, for example, ok, what does it look like in our area, or how does it look maybe in Sweden or in Malmö, or in different areas in Malmö, but that it's like, ok if you live in a certain area, maybe you can compare ok how do the emissions look there, compared to there. So that it's something they know that they can relate to."*

Estelle connected the actants "CME" and "real world". The connection between CME and the real world enabled her to use her pedagogical imagination to come up with a concrete example of what this would look like, i.e., examining emissions locally to connect to pupils' own real world. She also connected the actants "CME" and "pupils' age". This, we argue, imposes some constraints on her imagined situation in her pedagogical imagination process, in that she needs to find a way of working that suits the particular age group of the pupils.

Estelle continued by saying:

*"So I think it is, that everything can be learned but it has to happen early. It has to happen already at the first level, to bring in that thinking to see this abstract, to kind of see this critical and then it builds up, and you train."*

In her quest to achieve this imaginary situation, she addressed this potential age constraint by arguing that critical thinking must start at an early age, in order for pupils to learn what it is. Once again, we can see teacher education playing a vital role in addressing and clarifying such issues—in general terms, supporting affordance and removing constraints, if possible. She thought about how the diagrams look visually to the pupils, and she stated the following when comparing diagram 1 and diagram 2 (Figure 2):

*"The other feels a bit like -where are we. This is going to be ok. I think it will be clearer. It affects a lot. You want to capture interest, not lose interest in it."*

Furthermore, Estelle connected the actants "CME" and "fear and hope". The actant "fear and hope" imposes some constraints on Estelle's quest towards the imagined situation and constrains her in her pedagogical imagination because she is afraid that the pupils will lose interest if the diagrams are not clear enough. Her way of foregrounding her role as future teacher in relation to CME and climate change shows more of a *caring role*. She

thinks about the feelings that CME and climate change might stimulate, as well as which age group this teaching would work for and how it could be visually displayed so that it becomes easy for young pupils to understand.

Summarizing the discussion of the above cases of student teachers who participated in these interviews, we noted three points as we cross-examined them: (a) some actants are more frequently used than others in student teachers' attempts to foreground themselves as critical mathematics educators, (b) the quality of the actant networks created varies across the five cases, and (c) student teachers' sense of the complexity differs. First, as we can see in Table A2 in Appendix C, the two actants "CME" and "climate change" were the ones used most frequently throughout all five interviews. This has to do with the nature of the study, as CME and climate change were introduced in the statistics course and became a core part of the discussions. The student teachers referred to them and elaborated on them further, as we can see from their transcribed utterances. These elaborations can be seen in the actants created by the student teachers (i.e., "real world", "math as social activity", "instrumental statistics", or "manipulation") that show a relatively high frequency and can be perceived as crucial actants for the participants to foreground themselves as critical statistics teachers.

Second, the quality of the actant networks created by student teachers is also a significant outcome of this study. In particular, Nicole and Sophie created very simple networks without intermediate connections, while Nadir and Iman constructed much more complex networks with subnetwork relations, and Estelle's could be categorized as being of moderate complexity. An overview of the above case studies shows how the actant networks presented above illustrate how differently the student teachers conceptualize CME in the thematic context of climate change and how they foreground themselves as critical mathematics teachers. Specifically, all five were interested in CME in the thematic context of climate change, but two of them (Nadir and Iman) talked intensely and made intricate elaborations, while the rest were more reluctant to speak. The differences observed, of course, could be partially based on the personal histories of these five student teachers who volunteered to participate in the interviews. Despite their volunteer status, some were more reluctant to speak about what they were not sure about, while others gladly shared their thoughts. We could also add that there is more to explore with these interviewees than the actant networks presented here can show. Specifically, it was possible to see that giving a broader picture of each of the interviewees could have opened more dimensions in this discussion, such as the roles that could be played by other identity markers such as gender, race, or ethnicity. However, we also needed to follow through with our guarantee of total anonymity. This anonymity could also mean that the participants showed more willingness to speak more openly about their experiences on the course.

Third, the above allows us to more deeply grasp the complexity embedded in attempts to bring a CME perspective into school classrooms for discussing statistics through thematic contexts that deal with societal challenges such as climate change. We argue that the complexity increases in the sense that new actants and connections need to be introduced to the practicalities of designing teaching and learning in a CME context. The student teachers in this study indicated that this dimension cannot be ignored, but awareness is required. Iman's statement is indicative: "It is good to use statistics in different ways in teaching, you just have to think about not using everything at the same time, I think." There were also several occasions when the student teachers paused and made comments such as "I don't know what I am trying to say" and similar formulations.

Overall, all of the student teachers talked in positive terms with regard to using CME in the thematic context of climate change. The actant networks discussed here show similarities in that all of the participants brought in CME and climate change through their utterances as very positive things to incorporate. The noted differences across the actant networks produced concerns with respect to the number of actants that the student teachers connect with their foregrounds as critical statistics teachers. This could be due to either a reluctance to speak of what is not clear or the fact that they simply did not

see any other actants, i.e., ways forward. Since this study constitutes an introduction to CME in the thematic context of climate change, we would like to add that more actants would probably come into play as the student teachers progress in their understanding of and familiarity with CME. However, the student teachers' own hesitation is noted, either through articulating their opinions regarding the actants through the interviews and making specific actant connections, or through body language, silences, and pauses as the interview was carried out. Certainly, these issues need further research in the context of future studies.

## 5. Concluding Remarks

We agree with other colleagues in this Special Issue that critical mathematics education is an important opening for mathematics teaching and teacher education, for reasons already mentioned in the introductory section of this paper, and that its role is crucial for preparing students for a future as democratic citizens [37–40]. For this to happen in the context of teaching practice, teacher educators and student teachers must envisage themselves as future critical mathematics educators who are able to bring an active discussion of the formatting power of mathematics in society into the school classroom. In particular, the use of societal challenges such as climate change can be a thematic context for student teachers to enact critical pedagogical imagination, awareness, and sensitivity with respect to the embedded complexity. Moreover, critical mathematics education provides ways of enacting mathematical thinking grounded in cultural complexity [41] and becomes exemplified in cases where thematic contexts and contents reflect the desired culture of the curriculum, which risks ignoring the cultural diversity of the teaching and learning situation. The importance of CME when the thematic context of climate change is emphasized in teacher education was embraced by the student teachers in different ways. All student teachers foregrounded themselves as aiming to become critical mathematics teachers, but they expressed varied degrees of interest, doubt and hesitancy, and, sometimes, refusal. This became evident in the ways in which they resorted to different actants for articulating their foregrounds as critical mathematics teachers.

Bringing a CME perspective to how a statistics course embraces data related to questions of climate change inevitably increases the complexity of the teaching and learning process. We found that this increased complexity is perceived when new actants and connections are introduced by the student teachers themselves to justify their future choices. The actants introduced will, in turn, produce new actants that are grounded in the ways in which each of the student teachers conceptualizes CME in the thematic context of climate change, as we have seen from analyzing their interviews. At the same time, these new actants provide both constraints and affordances—for instance, when the student teachers started to talk about "source criticism" and then suggested ways of working with CME, such as looking for graphs in newspaper articles. Source criticism is a research field in itself and brings with it certain ways of doing things with data, truth, reliability, fake news [42], etc. For instance, investigating a homepage's network addresses the credibility of information [43]. As an example, looking at the homepage's top-level domain (i.e., .com, .gov, .edu) can give some indication of the credibility of the homepage [43].

We can see many similarities between our study and, for example, the studies by Steffensen [5,7] examining how participants articulated CME in the thematic context of climate change, but while she examined their attempts to articulate in relation to critical competencies, the ANT approach allows actants to reveal themselves through the student teachers' utterances. There are pros and cons to each approach, but we argue that an advantage of our approach is that it allows the actants themselves to reveal what was not yet known in advance, and this could be beneficial for teacher educators to tap into. For instance, the actant "teacher traditions" was an unexpected actant for us in this investigation. Here it is important to ask how student teachers relate to a normative position such as 'teacher traditions' in their attempts to open up for lessons that incorporate critical mathematics teaching and learning? This might include taking a political activist stance

that foregrounds a specific ideology or embracing a pluralistic approach allowing all types of opinions—even climate change denial. In turn, this dilemma might also suggest specific modes of doing things; for instance, what data to choose, what graphs to select and, overall, creating spaces in which student teachers foreground themselves as critical statistics teachers. We would like to add that we, in this text, ourselves as researchers did not try to teach the student teachers what to do, but instead, tried to learn with the student teachers in this complex process. When facing urgent societal challenges like climate change, some may opt toward an activist stance that strives to apply solutions for resolving the problems by avoiding engagement with a time-consuming democratic process. However, we argue that there is still a need to learn with the student teachers by opening up such complex discussions, despite the risks of not being able to produce always viable answers.

Creating more awareness of how such actants could potentially work so that teacher educators could grasp the complexity of bringing CME into their mathematics teacher education courses could be a proposal derived from this study. In relation to grasping the complexity, such actants could help them not only to reduce the constraints, but also to build and work with them as potential affordances. At the same time, this will also put more focus on the process of stimulating pedagogical imagination and bridging the current and the imagined situations, which we argue should be an important part of teacher education and is also emphasized by Skovsmose et al. (2023) [37] in this Special Issue.

The actant networks presented here, both in the five cases discussed in Section 4 and as generalized actants in Appendix B, can be used as exemplary vignettes for creating reflexive dialogues in mathematics teacher education courses or workshops. These vignette-based dialogues could invite student teachers to try to make their own actant networks with respect to how they foreground themselves as critical statistics teachers. This process would highlight the importance of pedagogical imagination to the student teachers. A similar proposal has been made by Rubel et al. (2021) [44], who discussed the critical reading of data visualizations by bringing CME into the thematic context of the coronavirus pandemic. They employed the triplet of *renarrating* (i.e., talking about what stories can be told in these pictures), *reframing* (i.e., what relationships should be highlighted), and *reformatting* (i.e., what has been left out).

We see our work as adding to such attempts to bring CME into mathematics education. Specifically, our actant network analysis brings an additional critical approach to critical mathematics education itself. Table A1 in Appendix B can act as a guide to potential actants coming into play during the design of teaching and learning that aim to use CME to discuss climate change. For instance, teachers could discuss how the actant "source criticism" is related to CME. Teachers could discuss the differences between "instrumental statistics" and "critical statistics" or consider how CME could be beneficial for democratic purposes, and even to describe what the United Nations' documents report about climate change. Moreover, to discuss how a student teacher should navigate the curriculum in relation to CME and climate change, one could give examples of how CME could become a social activity that brings joy and hope, rather than being frightening, and discuss what age group(s) this is suitable for and what different teaching traditions a student teacher could apply—being normative in driving an issue or taking a more "pluralist" approach, or even allowing the expression of opinions that come close to climate change denial [42] (p. 76). This approach requires an increased focus on the imagined situation, and that the mathematics teacher is prepared to consider other ideas than those described by the curriculum [37]. The complexity of some of the actant networks presented here might seem daunting and could cause reluctance to introduce CME among both teachers and student teachers. We see this study as an attempt to alleviate such hesitation. Having teacher educators who are somewhat familiar with this complexity might help reduce such hesitation. Another implication of the approach used in this study is that the teaching and learning situation will be co-constructed with the student teacher perspective in mind by the teacher educator. This would require that teacher educators learn to listen actively to

how student teachers engage with critical mathematics teaching and recognize the value of informal understanding as it is also highlighted by Hough and Solomon (2023) [45].

This study inquired into student teachers' foregrounds when introduced to CME through the climate change thematic context. We believe that further studies with student teachers who are much more familiar or knowledgeable with statistics, CME and climate change would be of equal interest. What would their networks look like and how would these compare with the ones created by the novices of this study? How do they envision themselves as critical mathematics teachers and how do their familiarity, knowledge, or experience with diverse concepts support their foregrounds? What different actants would this bring that are beneficial to teacher education? In a broader sense, this study shows how ANT can be used as a theoretical and methodological approach for studying how a particular teaching and learning setting (i.e., CME and climate change) could be enacted by student teachers as a process of pedagogical imagination in mathematics teacher education by appreciating its embedded complexity.

**Author Contributions:** Conceptualization, M.Ö., A.C. and L.B.B.; methodology, M.Ö., A.C. and L.B.B.; software, M.Ö.; validation, M.Ö., A.C. and L.B.B.; formal analysis, M.Ö., A.C. and L.B.B.; investigation, M.Ö.; resources, M.Ö. and L.B.B.; data curation, M.Ö.; writing—original draft preparation, M.Ö. and A.C.; writing—review and editing, M.Ö., A.C. and L.B.B.; visualization, M.Ö.; supervision, A.C. and L.B.B.; project administration, M.Ö., A.C. and L.B.B.; funding acquisition, M.Ö., A.C. and L.B.B. All authors have read and agreed to the published version of the manuscript.

**Funding:** This research was funded by the Swedish Research Council, grant number 2019-03679.

**Institutional Review Board Statement:** The leader of the Ethical Board of the University of Malmö considers that this study involves adults and since no obvious sensitive personal information will be handled, it is no need for an ethical application to the National Ethics Committee.

**Informed Consent Statement:** Informed consent was obtained from all subjects involved in the study.

**Data Availability Statement:** Data are contained within the article.

**Conflicts of Interest:** The authors declare no conflict of interest.

## Appendix A

Interview guide:

Advance information:

- Welcome and brief about myself (if necessary)
- Description of the interview
- Agreement on the planned duration of the interview
- Information on the use and publication of data
- Explanation of consent

Getting Started

A short introduction with a simple open-ended question to start the conversation.

Introduction with a brief description of the research and why I need the informant's help:

The research is about examining in practice how mathematics learning is affected by working with critical mathematics in mathematics teaching, with the help of examples regarding climate change. Critical math is illustrated in some examples that show that it is possible to create narratives with the help of data selection and choice of diagrams in statistics. The research aims to gain knowledge about this when teaching in teacher education (grades 4–6) in the subject of statistics. I need your help with experience from the course to be able to see what knowledge can be gained from this type of teaching.

Main part

Questions and follow-up questions:

Question 1: What did you think of the statistics course? (Simple, open question to start the conversation)

Supplementary questions:

Can you develop that?

If the discussion drives away from mathematics learning, some question related to this may be necessary. For example: I notice that you are very interested in the environmental issue, but how does it affect mathematics teaching?

If it comes to working methods, amount of work: How did you feel that the working method and the amount of work were in the work around critical math?

What do you see as other advantages and disadvantages of this way of working in mathematics teaching?

Question 2: In the lectures I mentioned the following examples. How did your thoughts go during the lecture and afterwards, around this example?

These statistics suggest that climate change must be addressed at country level.

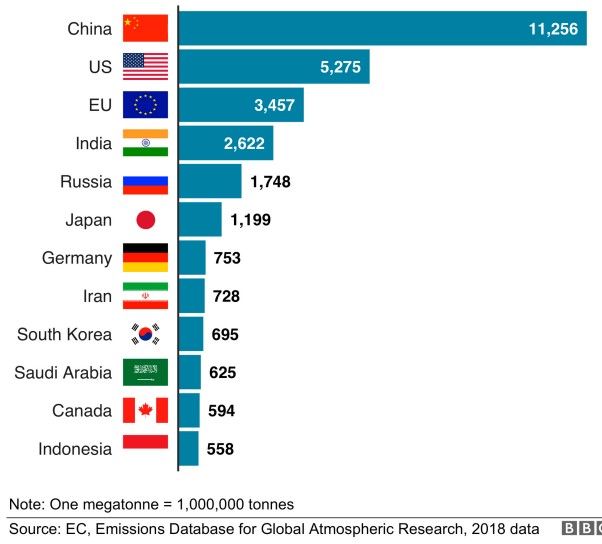

**Figure A1.** The world's to emitters of carbon dioxid.

When carbon dioxide emissions are presented below, it suggests that the discourse should be conducted as a global issue.

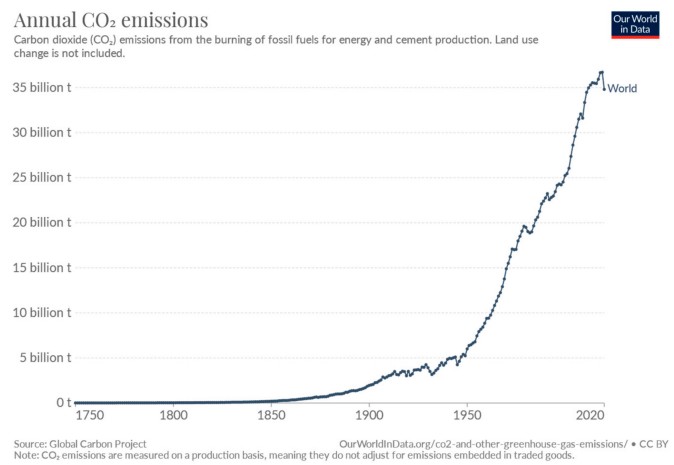

**Figure A2.** Annual carbon dioxide emissions in the world.

Supplementary questions:

If interest is shown in using this in their own teaching, appropriate questions may be: Do you see that the pupils should work with this themselves or is it the teacher who

should teach about it? Is it a collaboration between the pupils and the teacher or should the teacher inform about this? How do you see that the pupils can work with this themselves? What is the benefit of the teacher being aware of this (critical math)? Is there any benefit to mathematics teaching in that teachers and pupils work with statistics in this way? If so, which is it? How do you think it affects learning?

Question 3: Another example that illustrates how the choice of data and diagrams affects the discourse (Figure 2).

The diagrams shows carbon dioxide emissions for the United Kingdom and India, respectively. The first diagram shows how emissions for the two countries have changed since 1880.

The second is a commutative or accumulative diagram, i.e., the sum of the emissions can be seen on the right of the diagram.

The third diagram shows the emissions per person for the two countries between 1950 and 2008.

As can be seen, the different diagrams create two different images or narratives of the situation.

How could you as a teacher work with this in your teaching in a class, grades 4–6?

Follow-up questions: If the discussion turns to the issue of workload: Is there any way to weave this into the regular teaching so that the mathematical and critical math are treated simultaneously?

If the discussion drives too much to the climate issue, ask a question that brings you back to mathematics. (However, some thoughts on the climate issue are interesting to include). One question may be: I feel that you see many benefits regarding climate change as a subject, but how do you think it affects the learning of mathematics?

If the discussion is about concrete ways of working with this: Do you want to develop that? Again, if the answer is more about climate change try to link back to mathematics teaching.

Question (to show that I connect to what was mentioned previously): You mentioned before that... Do you see any connection between this and what you just mentioned?

Question 4: How would a class in grades 4–6 benefit from this in their teaching?

Follow-up questions: How do you think it affects mathematics teaching that awareness increases around these issues (critical math)? How did you, as a student-teacher, experience that it affected your own motivation and learning about working with statistics. How do you think it affects your motivation as a teacher?

Question 5: Is there anything else you want to add?

Review

Brief summary of the interview. In this way, misunderstandings can be avoided. Thanks for the interview.

Going forward

Description of what will happen next with the answers. Whether the interviewee will be informed about the continuation of the study and the results of the research.

**Appendix B**

Table A1. Description of the thematically grouped actants with example quotes from the interviews.

| Generalized Actant | Description | Example of Data Quotes |
|---|---|---|
| CME | This actant introduces the idea of critical mathematics education. The diagrams (Figure 2) format how the lectures will be carried out and imply certain ways of thinking. Opens new ways of thinking. | Sophie: "Yes, I think so, you get a more basic understanding of both the subject and the diagrams. If we hadn't had this discussion, we wouldn't have really understood what the diagrams are about. That is, what their underlying meaning really is. And that benefits learning, I think." |
| Democracy | Democracy as an actant refers to ideas about preparing citizens to be well informed and critical, e.g., valuable in democratic elections. Some STs referred to this as a reason for using CME. | Nadir: "Because if you present a table of that country and then it gets into perhaps society as well as within the social sciences. For example in China, it's not a, well, democracy in that way. Could that be the reason it looks, well, things like this. There are many reasons why for example." |
| Politics | The actant politics, in our case, refers to states of affairs that the ST cannot change or influence. Politics or similar words were used by some STs to talk about things that they felt that they had no way of changing or influencing. | Sophie: "Then you also know the level of politics that politics is different in the different countries. It can also affect emissions." Nadir: "it's not that easy to be able to link, we live in a political world, a lot lies in this kind of politics" Nadir: "Yes, well, politicians have a lot, really decide like this, this international, or this national, within our country, what shall apply, what is done." |
| Math as social activity | Involves discussion and acting out what is being discussed. | Nicole: " yes, that you think that mathematics belongs at school, it's not something I need to use at home. It's just something unnecessary that you have to do at school, but if you do it more socially, you might think that—well, it actually belongs outside as well. There is a reason why we study in school, it is so that we can use it in real life." |
| Instrumental statistics | Suggests a more traditional and instrumental way of looking at mathematics, i.e., technical, abstract, and disassociated from the social context of the data used. Some STs referred to previous experience of mathematics without CME. | Nicole: "You might get a different picture of mathematics, instead of that, this traditional mathematics teaching, that you just have to sit quietly and work in a book, that you get the whole thing, well mathematics can be a social thing, that you put in values, you can discuss, yes that you get a more enjoyable picture of mathematics as well." |
| Curriculum | Sets part of the stage for what should be done and what can be done during the lessons. Some STs referred to this in the interview as a constraint on what they are supposed to do in mathematics education. | Iman: "For one thing, the curriculum says that we must teach about the environment and the outside world and be able to reason and become a good citizen of society and make oneself understood." |

**Table A1.** *Cont.*

| Generalized Actant | Description | Example of Data Quotes |
|---|---|---|
| Pupils' age | This actant refers to ideas about how pupils' age puts some constraints on what can be taught at a specific age or maturity level. | Estelle: "It has to happen already at the first level, to bring in that thinking to see this abstract, to kind of see this critical and then it builds up, and you train, so that's why it's very late. It depends on who and what kind of prior knowledge they have with them." |
| Manipulation | The idea of not being told the truth. This can occur in a conscious or unconscious way. The idea that there is an objective truth, and that the teacher should tell and show that truth. | Nicole: "like yes you can manipulate quite a lot with diagrams depending on what you choose to focus on and what purpose you have with the various diagrams, sort of." Iman: "if you once show this and want to discuss how statistics can deceive the eye depending on how the bars are set" |
| United Nations (UN) | The UN provides guiding documents that the teacher relates to. This can be seen as one of the actants that build up the actant climate change. | Sophie: "I think the UN and their goals are very clear, [eum], then they have done this, that is, based on facts and research and all that kind of stuff. And then you can start from their ages, what do the students find interesting, what do they want to know." |
| Teaching traditions | Actant that refers to the idea about the teacher's role or what teaching tradition to follow—for instance, being normative and pushing a certain standpoint, or taking a "pluralistic" stance allowing all kinds of discussions. "Every tradition has its shortcomings and strengths." [46] (p. 76). | Iman: "Try not to put the words in their mouths, because then I only think certain will back off—"no, I don't want to" Iman: "Overcommitted if you stand very clearly on a position, politically for example, then it can be easy to push the issue." |
| Climate change | Actant that contributes as an important subject to illustrate CME and, at the same time, creates an awareness of the subject. | Iman: "The climate issue doesn't feel as problematic, if you do, because there you can stick to the facts quite a bit, I think. But that's because I think this is a big problem, but so do the majority of world leaders, even if not everyone agrees." Nicole: "right now we have a big climate crisis so we all really have to try to lower our emissions as much as possible so that it doesn't increase the greenhouse effect." |
| Source criticism | The idea that facts need to be checked for accuracy. | Nadir: "There is so much information online, but it must be from sources that are close to our time and so that you have discussions like this about source criticism." |

**Table A1.** *Cont*.

| Generalized Actant | Description | Example of Data Quotes |
|---|---|---|
| Real world | The idea that school is separate from the real world. Some STs mentioned that school activities need to connect to the real world, or words to that effect. | Sophie: "It is very easy to believe that it is only like this. But it is not, there are many reasons why it is represented in this way. So their feeling towards mathematics, yes it gives them a more realistic and reality based picture. Can I imagine because math is abstract it needs to be more concrete that way." Nadir: "but how to make it clear to students and that they shall find this connection between then, as I said, reality and these tables and diagrams." Nicole: "well, it actually belongs outside as well. There is a reason why we study in school, it is so that we can use it in real life." Nicole: "It is quite often that you get the question, well, why should we learn this here? Why should we learn to calculate the root of something, we don't need to be able to do that just like in reality. But sometimes you actually need to be able to do that, as well" |
| Fear and hope | This actant has to do with STs' concerns about pupils' feelings. Climate change can be frightening. Some STs expressed concern about the feelings that a certain subject might inflict on their pupils. | Sophie: "a little like this that the students get this type of negative image like this, like that the topic that you need to touch on and, especially for the younger ages, because we have talked about that in other courses that it should not be too negative so that, that the students get too negative a view, there is hope there is like, present it that way, but at the same time they need to be aware of this stuff." |
| Interdisciplinary | The idea of introducing a combination of multiple academic disciplines into one activity—for instance, mathematics and geography. | Iman: "I am very much in favor of doing it at the same time. I like the integrated subject. So I absolutely think that this could be brought up in both the NO lesson and in the math lesson." |
| Social justice | Actant that contributes as an important subject to illustrate CME and, at the same time, creates an awareness of the subject. | Sophie: "Then equality and the social problems are also a big issue. But I would probably have chosen the climate issue first." |

## Appendix C

**Table A2.** Frequency of each of the generalized actants used in all interviews; the top row shows generalized actants, while the first column shows the participants.

| | CME | Democracy | Politics | Math as Social Activity | Instrumental Statistics | Curriculum | Pupils Age | Manipulation | United Nations (UN) | Teaching Traditions | Climate Change | Source Criticism | Real World | Fear and Hope | Interdiciplinary | Social Justice |
|---|---|---|---|---|---|---|---|---|---|---|---|---|---|---|---|---|
| Nicole | 5 | | | 4 | | | | 2 | | | 1 | | 3 | 1 | | |
| Sophie | 11 | | 3 | 2 | | | 1 | | | | 11 | 2 | | | | 3 |
| Nadir | 11 | 2 | 2 | 3 | 4 | | | 2 | 2 | | 6 | 6 | 5 | 1 | 2 | 1 |
| Iman | 10 | | | 2 | 3 | 2 | | 4 | | 2 | 4 | | 2 | 1 | 1 | 2 |
| Estelle | 14 | | | 3 | 4 | | 5 | 4 | | | 1 | | 5 | 3 | | |
| Frequency | 51 | 2 | 5 | 14 | 11 | 2 | 8 | 10 | 2 | 2 | 23 | 8 | 15 | 6 | 3 | 6 |

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
