# Peer review of "Bringing Critical Mathematics Education and Actor–Network Theory to a Statistics Course in Mathematics Teacher Education: Actants for Articulating Complexity in Student Teachers’ Foregrounds"

_education, doi:10.3390/educsci13121201_

Round 1
Reviewer 1 Report
Comments and Suggestions for Authors
This paper should eventually be published in this special issue. It contributes important applications of theory to important perspectives on practice in teacher education. It is a model for others. However, the English writing of almost every sentence needs editing by a native speaker. I was able to read through the paper only because I know the theories and the issues/questions/ideas very well myself. It was intensely frustrating to try to read the manuscript carefully. Nevertheless, I managed, because the ideas in the paper deserve my attention.
I also have some suggestions about organization:
The opening section spends far too much time on Skovsmose. I too am a fan of his. However, you are not Ole's manager/promoter: you are another researcher doing your own work. Such detail reads like the literature review of a doctoral dissertation, rather than the introduction to your own article. What is the way that a reader can be introduced to YOUR work, the basic idea, and its importance for others? That should be learned through reading your first few paragraphs, rather than a summary of Ole Skovsmose's own work.
Why, do you think, only 5 student teachers agreed to be interviewed? The power-knowledge dynamics of this should be part of your analysis of the complexity of ANT. The researcher/teacher-educators are also actants in this research, and impact on the choices and decisions of the potentially larger number of interview informants. Statistics and climate change and CME are all actants as well. My thoughts are that your analysis should be developed much further with all of this in mind. Otherwise, we only get a 1-dimensional perspective on the complexity going on.
Finally, if you reduce the open sections substantially, the actual discussion of the research itself, the data and its analysis, and then the conclusions, are all far too brief. Especially lacking is a discussion of the implications for teacher education. All we get in the end is a vague appreciation for the complexity that is now thrown at the student teachers, rather than suggestions for practice and further research. I would especially like to know more about the foreground/background of those interviewed: did they bring aspirations and expectations about teaching, climate change and statistics to this experience, and do they now have similar/different relationships with one or more of these actants?
Are youre recommendations really only reduced to inviting discussion into the classroom? For an international audience, this will seem ridiculous, since current professional principles for mathematics teaching in most Anglo-influenced countries, at least, have expected discussion to be part of instruction for several decades. You do mention five foregrounds: I hope a revision of this paper will elaborate applications of the data analysis for each of the five of these. There is some of this potentially included in the table in Appendix B; a summary of this table in the body of the paper would help, along with further discussion.
Comments on the Quality of English LanguageThe English writing of almost every sentence needs editing by a native speaker. I was able to read through the paper only because I know the theories and the issues/questions/ideas very well myself. It was intensely frustrating to try to read the manuscript carefully. Nevertheless, I managed, because the ideas in the paper deserve my attention.
Reviewer 2 Report
Comments and Suggestions for Authors
This paper is compelling and thought-provoking, particularly in its introduction of theoretical perspectives to illuminate students' imaginations and understanding of CME within the context of climate change. I find it intellectually stimulating, especially when reading about the potential contributions of Actor-Network Theory (ANT). Nevertheless, to prepare this paper for publication in the special issue, there are several areas that should be addressed:
- The brief statement of purpose could benefit from the inclusion of research evidence at the outset of the manuscript.
- The contributions of ANT should be more thoroughly elaborated upon when it is first introduced in the paper to make a more compelling and prominent case for its valuable contribution. Consider utilizing content from lines 120-130 to achieve this.
- Additional sub-headings in the methodology section would enhance clarity.
- It is unclear who conducted the interviews. If the first author, who designed the course, was responsible, it would be beneficial to include some reflexive comments on potential conflicts of interest during participant selection, interviews, and data analysis.
- It would be helpful to provide an example of a data analysis extract accompanied by the relevant theoretical concepts.
- Ethical considerations section should be added and the approval from the relevant committee should be included.
- Could you specify the digital tool you used to illustrate the connections between the actants?
- It seems that the cases of student teachers represent the findings of this paper. It would be helpful to signpost this after the data analysis section, possibly with a different level of subheading.
- After a comprehensive theoretical background section, the findings don't seem to fully leverage the strong and valuable theoretical framework. This issue can be addressed by incorporating key concepts mentioned within the cases and elaborating more on the cases themselves. Consider discussing any divergences or convergences between the cases and possible reasons for them.
- The statement, "it will just be a pancake if you try to lecture about something you don't know," may hold specific cultural relevance but could be unclear to readers from other backgrounds. You may need to provide additional context or consider rephrasing for a broader audience.
- Quotations should be made more easily identifiable within the manuscript, perhaps through the use of italics or indentation.
- How did the content of the module influence the responses of student teachers?
- Similar to the findings section, the conclusion would benefit from strengthening by incorporating references to relevant literature.
Reviewer 3 Report
Comments and Suggestions for Authors
The article Bringing CME and ANT to a statistics course for mathematics teacher education: Actants influencing student-teachers’ foregrounds for classroom teaching addresses an interesting and compelling topic: the complexity that in-service student teachers in a Swedish teacher education program encounter when they are introduced to critical mathematics education (CME) as an approach to understand climate change more nuanced through statistics, and how this experience may influence their prospective teaching in mathematics.
More specifically, the article explores what networks of relations that may transform the student teachers’ attempts to foreground classroom teaching within the milieu of CME through the thematic context of climate change. The article presents data from a study that consist of a statistics course for student teachers and individual interviews with students from the course (N=5), conducted after the course. The authors draw on the perspective of actor network theory (ANT) in their analyses of interview data and discussion of findings.
The article begins by highlighting the complexity student teachers encounter when CME is applied in their statistics course in combination with climate change and how this influence their imagined future teaching mathematics. The article highlights the potential of CME to empower people through critical thinking (rather than people perceiving mathematics and statistics as neutral or objective and thereby more likely to become manipulated). This issue is revisited and addressed towards the end of the article. The authors argue that the “complexity increases, compared to more traditional instrumental statistics, when a student-teacher moves into teaching by encountering CME and climate change”.
Thank you for interesting reading! In my view, the article has several strengths. It is conceptually interesting. As mentioned, the authors drew on the perspective of CME in their overall framing of the article, in their analyses of data, and in discussion of findings. I felt that this conceptual construct was highly relevant to the research presented in the article and to the call of the special issue.
Furthermore, I think that the article raises an extremely important argument about how CME can empower (student) teachers as they grapple with how to educate future generations of pupils to better understand and handle issues related to the climate crisis through a critical perception of statistics, including how statistics may be used to support various (biased) arguments in political debate on climate change.
However, I do have one major concern: that it is hard for me to follow the argument, because I am not sure that I thoroughly understand what this complexity is about and how it may contribute to empower student teachers. This might be because the complexity is referred to in different ways throughout the article and because some information in the article remains implicit. The article would benefit from clarification of the research context, theoretical concepts and method.
I appreciate the complexity of bringing together, multiple amorphous concepts, such as CME, ANT, actors, (generalised) actants, agency, foreground, background, and CS, AS, IS etc. However, I think most of these concepts need further development in order to appropriately understand how they are being employed in this article. For instance, I do not fully understand what qualifies for a human or non-human action to be an actant? How did you decide what actants are important in the analyses? Coding in data analysis? Frequency in interviews? Did you triangulate to identify overlaps between observation (“teacher’s log”) in the course and the interview data, or something else?
I invite the authors to consider the following revisions to strengthen the article:
The complexity: The reader needs to better understand what the complexity is about. Is it about societal and institutional hidden agendas that constraint CME? (page 2, lines 69-72). Is it that public debate about climate changes may use statistics in biased ways? Is it about students engaging with climate change as a process, not a fixed outcome? (page 7). Is it the bringing together of CME and climate change in education that creates complexity, as it becomes more challenging for the students to envision the future teaching? (page 16). Please consider how you could unfold this complexity more thoroughly.
For instance, it might help to unfold the findings of Vithal (2000) to clarify this issue. What issues and tensions does she explain? Maybe going more into the formatting power of mathematics in the analyses could help the reader to better understand. Also, clarification of theory and method could enhance coherence in the article, as suggested below.
Title: Many readers are not familiar with the abbreviations CME and ANT. You might want to attract international readers by making a more appealing title that addresses the theme of the article in more overall and general terms. You could, for instance, use key words and a quote from interview data so the title reflects the argument you wish to make, such as “Those who go to school now will be future decision makers”.
Introduction: I suggest that the authors begin by introducing the overall problem or potential (the empowering potential of CME) addressed in the article, in the societal context. The authors do not write what country the study takes place in, but I guess that it is Sweden? The Swedish curriculum is mentioned, and there are references to Swedish language and areas in the findings section. Please write up front what country the study takes place in.
Are there anything in the Swedish societal context that makes this research particularly relevant? Might the national curriculum for mathematics education constraint or enhance CME – just wandering? As it appears CME seems to align w the general Swedish curriculum? Curriculum is addressed in some of the analyses, but I did not entirely understand the implications – maybe because I need to understand the context of the study better.
Do you need two research questions, if you do not answer the first one, and the second one is an elaboration of the first one? Maybe you just want to state the aim of the article in the introduction?
Theoretical considerations: As mentioned, I appreciate the complexity of bringing together, multiple related concepts, such as CME, ANT, actors, actants, agency etc. I understand CME, but I think that the rest of these notions need further development in order for the reader to appropriately understand how they are being employed in this article. For instance, I do not fully understand what qualifies for a human or non-human action to be a generalized actant?
You write that you consider the students as co-researchers. This could be placed and unfolded under the method section.
Do lines 260-264 belong under method? Here you explain how you constructed the interview questions.
Methodology: It is interesting and relevant that the authors use data from a course and interviews. Yet, the methodology section needs to be thoroughly revised to enable the reader to understand what this study is about. I am curious to know what the overall conceptual approach of the study is? Is it a case study, some kind of action or intervention-inspired study? You mention self-study (page 7) under method. Please provide a reference for somebody else than the authors or/and explain what this approach is about. The theory section mentions that the students are co-researchers, and the findings analyse individual case profiles... If you go more into case study, you might find the research of Bent Flyvbjerg inspiring. Or maybe you just want to foreground ANT (Järvinen and Mik-Meyer) as the overall approach?
Please be more specific about what data you include in the study and how you collected these data. Is the course part of data? You mention the teacher’s log. What is this? Does the article need to include the diagrams under method, or might they be in appendixes? Maybe the authors just want to explain how the students acquired critical competences in understanding how different diagram types changes the perception of a given phenomenon?
Please include some consideration about ethics. Sweden is one of the strictest countries in the world, when it comes to research ethics. What actions have the authors conducted to ensure research ethics? Has the study been approved by the Swedish ethics authorities? The first author teaches the participants of the study. What ethical issues might raise from that? What did you do to handle such issues, or why is this not an ethical problem?
I believe the sentences about work distribution between the authors (page 7, 332-333) belong in an author statement.
For inspiration to revise methodology, see the methodology section of this article (Solomon et al., 2023): https://doi.org/10.3390/educsci13090960
Findings: The sections on how data was collected and analysed data (page 11, 470-505) belongs under methodology.
As mentioned, I struggled to thoroughly understand the analyses. Reading appendix B was a great help. The appendix overviews some very interesting interview quotes. The question is whether such essential information regarding the analyses belongs in an appendix or should be included in the findings? I think it would help the reader understand the analyses better if you found a way to include the identification and exemplification of actants (appendix B) in the findings section.
Also, please be more specific in treating the data. For instance, page 21 says “ST refer to this in the interviews…”. How many students do refer? Being more specific could help the reader better understand what qualify as an (generalized) actant.
The term “infralanguage” is mentioned for the first time in the article in findings. Is this part of the conceptual framework? Then please explain in previous sections.
Comments on the Quality of English LanguageThe general quality of English is good. However, an idea would be formulate more straight forward and short sentences. This would make the language more clear.
Round 2
Reviewer 1 Report
Comments and Suggestions for Authors
The revisions to the original manuscript have made a huge difference in the quality of this submission. The article in my mind makes a powerful contribution to the field, beyond statistics and social contexts of mathematics teaching and learning to the complexity of teacher education as a form of dispositional transformation.
The additional material supports the themes and argument(s) that the authors present, and make it clearer that their own work (more than those publications that inspired their work) is important and worthy of many readers' attention.
I wish they would develop more recommendations and implications of this study for how to create teacher education experiences that facilitate richer and deeper understandings of the 'problems' with the forms of teaching and learning that led them to want to be one of those teachers that are implicitly criticized by this research. How complicated is that!??!? All of the motivations for becoming a teacher (or at least a lot of them??) are ... "wrong" ?
Also: is the study of climate change enough to save our planet? Maybe prospective teachers need to abandon their career goals in favor of more direct forms of social and political activism? This issue is not addressed in this manuscript, yet might be more significant than good kinds of teacher education?
I recommend "accept" yet hope that the authors might consider one more round of minor revisions that address the questions I have posed here.
Comments on the Quality of English LanguageThe English in this version is vastly improved, and flows very nicely. There are still a few grammar errors scattered throughout.
Reviewer 2 Report
Comments and Suggestions for Authors
Thanks to authors for addressing every single point very clearly and effectively. I think the paper is now in a very high quality to be published in the special issue.
Author Response
Thank you for your kind words. The final version has now been submitted.
Best regards
Reviewer 3 Report
Comments and Suggestions for Authors
Thank you for the opportunity to read this greatly improved manuscript! I have only one suggestion for a minor addition; that you mention up front that the study takes place in Sweden, e.g., in page 1, line 36 you might add "at a teacher education program in Sweden" to the sentences that talks about the research site.
